# Sub-Nanometer Acoustic Vibration Sensing Using a Tapered-Tip Optical Fiber Microcantilever

**DOI:** 10.3390/s23020924

**Published:** 2023-01-13

**Authors:** Chunyu Lu, Mahdi Mozdoor Dashtabi, Hamed Nikbakht, Mohammad Talebi Khoshmehr, B. Imran Akca

**Affiliations:** LaserLab, Department of Physics and Astronomy, VU University, De Boelelaan 1081, 1081 HV Amsterdam, The Netherlands

**Keywords:** acoustic vibration sensor, microcantilever, tapered-tip fiber

## Abstract

We demonstrate a highly sensitive acoustic vibration sensor based on a tapered-tip optical fiber acting as a microcantilever. The tapered-tip fiber has a unique output profile that exhibits a circular fringe pattern, whose distribution is highly sensitive to the vibration of the fiber tip. A piezo transducer is used for the acoustic excitation of the fiber microcantilever, which results in a periodic bending of the tip and thereby a significant output power modulation. Using a multimode readout fiber connected to an electric spectrum analyzer, we measured the amplitude of these power modulations over the 10–50 kHz range and observed resonances over certain frequency ranges. Two types of tapered-tip fibers were fabricated with diameter values of 1.5 µm and 1.8 µm and their frequency responses were compared with a non-tapered fiber tip. Thanks to the resonance effect as well as the sensitive fringe pattern of the tapered-tip fibers, the limit of detection and the sensitivity of the fiber sensor were obtained as 0.1 nm and 15.7 V/nm, respectively, which were significantly better than the values obtained with the non-tapered fiber tip (i.e., 1.1 nm and 0.12 V/nm, respectively). The sensor is highly sensitive, easy to fabricate, low-cost, and can detect sub-nanometer displacements, which makes it a promising tool for vibration sensing, particularly in the photoacoustic sensing of greenhouse gases.

## 1. Introduction

The sensitive detection of acoustic waves has paramount importance in various applications ranging from health care to environmental monitoring [1,2,3,4,5,6]. In the past decade, this field has gained significant momentum, specifically in monitoring atmospheric greenhouse gases. While there are different approaches to gas detection, existing solutions all suffer from the same problems: (1) They are costly; (2) depend on a power source (such as batteries); (3) require manpower for regular maintenance; and (4) suffer from a tradeoff between sensitivity and coverage [7,8,9,10,11]. Additionally, they are not multiplexed nor do they simultaneously offer multiple gas sensing. Recently, IBM (International Business Machines Corporation, Armonk, NY, USA) has developed a photonic integrated system based on the absorption spectroscopy technique [9]. Each head incorporates its optics and electronics in one chip, which is then positioned in the point of interest. The system is thus not multiplexed and it requires calibration as it is based on absorption spectroscopy. Photoacoustic spectroscopy (PAS) is a well-established technique for gas trace detection [12]. PAS has a detection sensitivity for the concentration of trace gases of up to parts per billion and has a high specificity enabling the detection of individual gas species [13,14]. The PA effect is the process of sound generation in a material resulting from the absorption of photons. Several acoustic vibration sensors have been developed in the past with varying figures of merits [3,4,5,6,12,15,16,17,18]. Among them, optical fiber-based solutions have drawn increasing attention due to their tolerance to electromagnetic interference, small size, low weight, and ability to multiplex with a single sensor probe across a long distance [15,16,17,18,19,20,21]. Recently, an optical fiber-based sensor called PAS-WRAP was developed, which uses the PA effect for methane leak detection [22]. Although this is a very promising technology, the current version has limited detection sensitivity mainly due to the insufficient acoustic response of the regular fibers. Optical fiber microcantilevers can be used as an alternative to regular fibers since they can reach ultrahigh sensitivity in the detection of small signals such as mass, forces, chemicals, and biological species [23,24,25,26,27,28,29]. The working principle is based on monitoring the power variations caused by the misalignment between the pre-aligned cantilever and readout component. However, to enable high accuracy and high sensitivity, high-precision fabrication platforms as well as crucial optical alignments are needed, which lead to exceedingly complicated sensing devices and expensive fabrication equipment. Therefore, new concepts exploiting the benefit of fiber cantilevers are strongly needed.

In this work, we developed a highly sensitive and easy-to-fabricate tapered-tip optical fiber microcantilever that can detect sub-nanometer acoustic vibrations (Figure 1a). The tapered-tip fiber has a unique output pattern, which is ultra-sensitive to acoustic vibrations. Using a transmission-based measurement method, we detected the natural resonance frequencies of these fibers in the frequency range of 1–50 kHz. Two fibers with different tip diameter values were fabricated using single-step chemical etching and their performance was compared with a regular untampered fiber. A maximum sensitivity value of 15.7 V/nm and a minimum detection limit of 0.1 nm were measured for the fiber microcantilever with a tapered tip diameter of *d_t_* = 1.8 µm at 18.2 kHz, whereas the sensitivity and the detection limit of the non-tapered fiber were 0.12 V/nm and 1.1 nm, respectively. This sensor is ultra-sensitive, low-cost, and has a sub-nanometer limit of detection, which makes it a very promising tool for the on-site photoacoustic detection of greenhouse gases.

## 2. Materials and Methods

### 2.1. Working Principle

The number of modes of a step-index fiber is given as:(1)$$M=\frac{V^{2}}{2}=2{\left(\frac{\pi d}{\lambda }\right)}^{2}(n_{c}{}^{2}-n_{cl}{}^{2}),$$
where *d* is the diameter of the fiber, and *n_cl_* and *n_c_* are the refractive indices of the cladding and the core, respectively. Our fiber has no cladding layer, which can support ~4583 modes when *λ* = 660 nm, *d* = 9 µm, *n_c_* = 1.5, and *n_cl_* = 1. We continuously tapered the core of the uncladded multimode step-index fibers (SMF-28) down to a certain diameter (*d_t_* = 0.5 µm, trapezoidal shape), which forces the higher-order modes to radiate to the surrounding medium at different points along the taper with a certain angle. The radiated modes interfere and form a circular fringe pattern (Figure 1c) [30]. When an acoustic field is applied to the fiber tip, the tip starts bending periodically, which changes the refraction angle of the higher-order modes and results in a different fringe pattern (Figure 1d). The change in the fringe pattern dramatically affects the output signal read by the multimode fiber. At the resonance frequencies of the tapered-tip fiber, the change in the signal amplitude is amplified, which results in much higher sensitivity values in contrast to non-resonant frequencies. In this concept, the tapered-tip fiber acts like a microcantilever, which is a suspended micro-scale structure supported at one end, which can bend and/or vibrate when subjected to a load. In our case, the tapered-tip fiber is the micro-scale beam structure, which guides the light itself rather than reflecting an external beam in the case of traditional microcantilevers. Acoustic waves generated by the piezo transducer were the loads that we imposed on this structure. The tapered-tip fiber was fixed on the non-tapered side to provide support.

The displacement values for each piezo voltage were extracted from a Fabry Perot setup in which a regular fiber was pointed at the piezo surface and the facet reflections and the reflections from the piezo surface were used to form a Fabry Perot cavity. The experiments were repeated three times and we obtained a linear relationship between the piezo voltage and displacement, as shown in Figure 2. The inset shows the physical meaning of the measured displacement.

### 2.2. Fiber Fabrication

Tapered fiber tips were fabricated using a single-step wet etching process under controlled conditions. A simple etching setup, which consists of an XYZ stage with a fiber holder attached to it, was developed to obtain reproducible tapers of the desired diameter and length. First, the plastic coating around the fibers was stripped off mechanically and dipped into the 48% HF acid solution perpendicularly. The time of etching defines the diameter of the fiber tip. For the diameter of 1.5 µm, the etch time was 40 min whereas the diameter of 1.8 µm was achieved in 35 min. The same type of optical fibers with a core diameter of 9 µm was used in the experiments. The HF acid forms a concentration gradient (higher at the bottom and lower at the surface) that results in a faster etching rate at the fiber tip [31]. As a result, the fiber tip has a trapezoid shape. An optical microscope image of the fabricated fiber tips is shown in Figure 1b. The monitoring of the fiber diameter was conducted by observing the circular fringe patterns with a light source centered at 632 nm. We found that the diameter of the fiber decreased linearly with time with hydrofluoric acid used as an etchant at room temperature. The observed etching rate was 0.005 ± 0.0002 s^−1^, which was repeatable using the procedure developed in this study.

The length of the tapered part was ~10 mm in both cases. The tapered fiber length is a compromise between its sensitivity, robustness, and resonance frequency range. Shorter tips resonate at higher frequencies, which can be a limiting factor for the experimental part. Moreover, the sensitivity of the fiber is reduced for shorter tips. The length we chose in this study provides the best sensitivity that can be detected with the available equipment. The fringe pattern depends on the fiber length and fiber tip diameter; however, since our fibers were relatively long (millimeter range), the tip diameter had the biggest effect on it.

### 2.3. Measurement Setup

The setup depicted in Figure 3 was used to measure the deflection of the fiber microcantilevers for sound waves at different frequencies and amplitudes. The sound waves were generated using a low-voltage piezo element (Thorlabs, Newton, NJ, USA, PA1CEW), which was glued to an aluminum block. An arbitrary waveform generator (KEITHLEY 3390, resolution of 0.1 mV) was used to drive the piezo element. The tapered fiber was glued on a vertical translation stage (Thorlabs, VAP10/m) to control the distance between the fiber and the piezo element. A single-wavelength laser (Thorlabs, Pro800) was directly connected to the tapered-tip fiber and a 50-µm-core multimode fiber (M42L01, Thorlabs) was used on the detection side. The multimode fiber had a numerical aperture of 0.22, a core diameter of 50 ± 1 µm, and a cladding diameter of 125 ± 10µm. The light received by the multimode fiber was measured by an FC/PC-coupled photodetector (Thorlabs, DET08CFC/M), which was connected to an electric spectrum analyzer (ESA, Signal Hound, SA44B). A custom Labview program was used to control sound generation and data acquisition. The measurement was carried out by sweeping the frequency with a step of 50 Hz over the frequency range of 10–50 kHz. The ESA settings were as follows: a span of 100 Hz, a resolution bandwidth of 10 Hz, a video bandwidth of 10 Hz, and a reference level of −30 dBm.

### 2.4. Eigenfrequency Analysis

The eigenfrequency simulations were performed using the finite element method (FEM, COMSOL Multiphysics). In the simulations, an optical fiber with a core diameter of 9 µm was tapered to *d_t_* = 1.5 and 1.8 µm over ~10 mm length. The frequency range was 5 to 50 kHz. The density, Young’s modulus, and Poisson’s ratio of the fibers were 2203 kg/m^3^, 73.1 GPa, and 0.17, respectively. The simulation results are given in Table 1. For each fiber tip, several eigenfrequencies were obtained within the given frequency range. The strength of each eigenfrequency cannot be extracted from FEM simulations; therefore, it may not be possible to excite all these modes in the experiments. For larger taper tips, the resonance frequencies shift toward smaller values and the number of resonances increases.

## 3. Experimental Results

### 3.1. Measurements with a Tapered-Tip Diameter of 1.5 µm

We measured the frequency response of the tapered-tip fiber with a tip diameter of *d_t_* = 1.5 µm, which is given in Figure 4a. Two strong resonances were observed at *f*_1_ = 26.2 kHz and *f*_2_ = 41.1 kHz, which are in accordance with the eigenfrequency analysis results. The limit of detection (LOD) and the sensitivity values of 0.1 nm and 9.9 V/nm were obtained, respectively, at *f*_1_ = 26.2 kHz. At *f*_2_ = 41.1 kHz, these values were reduced to 0.2 nm and 3.2 V/nm, respectively (Figure 4b,c). The reason is that the resonance peak at *f*_1_ = 26.2 kHz has a higher amplitude compared to the resonance peak at *f*_1_ = 41.1 kHz. Even though we observed 12 eigenfrequencies for this tapered-tip fiber, we only observed two of them. One possible reason could be that the fiber tip is not big enough to experience a strong acoustic field for exciting all resonances. The ESA output at three piezo voltage values, *V* = 0.1 V, 5 V, and 10 V for *f*_1_ = 26.2 kHz, are given in Figure 4d. To determine the LOD values, the threshold amplitude for the minimum detectable signal was set to 10 dBm. As can be seen from Figure 4d, the signal-to-noise ratio (SNR) at *V* = 0.1 V was ~10 dBm, and therefore for this frequency, the LOD value was determined as 0.1 nm.

### 3.2. Measurements with a Tapered-Tip Diameter of 1.8 µm

The frequency response of the tapered-tip fiber with a tip diameter of *d_t_* = 1.8 µm was measured for different displacement values, which is given in Figure 5a. Within the given frequency range, several resonance frequencies were observed, as predicted by the eigenfrequency analysis. The LOD and sensitivity values of the microcantilever sensor were measured at the first two strong resonance frequencies of *f*_1_ = 15.7 kHz, *f*_2_ = 18.2 kHz as 0.1 nm, and 12 V/nm, and 0.1 nm and 15.7 V/nm, respectively (Figure 5b,c). These resonance frequencies were very close to the simulated values of 15 kHz and 18.1 kHz. Their amplitudes were similar and their full width at half-maximum (FWHM) values were smaller compared to other resonances, which led to higher quality factors and thereby better LOD values. The frequencies of the other resonances with smaller amplitudes also had very good overlap with the simulated values. The ESA output at three piezo voltage values, *V* = 0.1 V, 5 V, and 10 V for *f*_1_ = 15.7 kHz are given in Figure 5d. At *V* = 0.1 V, the SNR of the resonance peak was ~12 dBm; therefore, the LOD of this frequency was determined as 0.1 V.

### 3.3. Measurements with Non-Tapered Fibers

To estimate the performance improvement provided by the tapered-tip sensor, we compared the results with a non-tapered optical fiber (*d* = 125 µm) using the same measurement setup. No resonance was observed in the 10–50 kHz range, as shown in Figure 6a. One reason is that the fiber was not sensitive enough to vibrate at the acoustic pressures that we used. The second reason could be that the fiber with a cladding layer is stiffer, and therefore resonates at higher frequencies. The inset shows the ESA output response at *f* = 24.6 kHz, for the piezo voltage of 12 V. The detection limit of the fiber was obtained as 7 nm, as given in Figure 6b. The sensitivity was estimated as 0.12 V/nm, which was >2 orders of magnitude smaller than the tapered-tip sensors. The large enhancement obtained by the tapered-tip fiber microcantilevers is mainly due to two main reasons: (1) the unique circular fringe pattern, which is ultra-sensitive to displacement, and (2) resonance-enabled signal amplification. The rigid fibers become very flexible after chemical etching. Due to this fact, they are not as fragile as non-etched fibers and they can bend relatively easily without breaking in contrast to non-etched fibers.

### 3.4. Effect of Fiber to Fiber and Fiber to Piezo Distances

We investigated the effect of distance (*l*) between the tapered-tip fiber and the multimode fiber as well as between the piezo transducer and the tapered-tip fiber (*g*) (Figure 7a). Even though it is common sense to expect higher SNR values when both *l* and *g* become smaller, it is important to check whether the number of resonances as well as the vibration sensitivity change with increasing distances. For instance, when *g* becomes smaller, the area of the fiber that is exposed to the acoustic pressure also becomes smaller. Although this can increase the strength of the existing resonances, it may also change the number of resonances that can be excited. When *l* becomes smaller, the amount of signal that is collected by the multimode fiber increases; however, the sensitivity of the fiber decreases by ~70–80% since the fringe pattern becomes compressed and the envelope of the measured signal has a Gaussian-like profile (as in the case of a non-etched fiber).

In both experiments, we used the tapered-tip fiber with a tip diameter of *d_t_* = 1.8 µm. The frequency response of the tapered-tip fiber was measured at three different distance values both between the tapered-tip and the multimode fiber (*l*_1_ = 0.5 mm, *l*_2_ = 0.85 mm, *l*_3_ = 1.2 mm) and the tapered-tip fiber and the piezo transducer (*g*_1_ = 0.1 mm, *g*_2_ = 0.6 mm, *g*_3_ = 1 mm), as shown in Figure 7b,c, respectively. The number of the resonances and their frequencies did not change in both cases, but the amplitudes of the resonances decreased with increasing distance, as expected. Based on these results, in all of our previous measurements, we used *l*_1_ and *g*_1_ values in the setup.

## 4. Discussion

The spectra we obtained from tapered-tip optical microcantilevers showed different resonances. The number of resonances increased when the tip diameter was *d_t_* = 1.8 µm. This can be explained by the fact that a thicker fiber has a larger volume to be seen by the acoustic field, and therefore it becomes easier to excite more modes. Moreover, the amplitude of the resonances also increases, which increases the sensor sensitivity. However, with an increasing tip diameter, the circular fringe pattern disappears, and eventually, a Gaussian profile is obtained, which decreases the sensor sensitivity (e.g., non-tapered fiber case). Therefore, there is a trade-off between the tip diameter and sensor sensitivity; thin tips have less volume for efficient acoustic excitation whereas thick tips have a Gaussian-like output profile.

In acoustic vibration sensing, it is important to find the right boundary conditions that provide repeatable results. Here, we fixed the tapered-tip fibers on a heavy translation stage using mechanically stable glue. Moreover, the length of the fiber that was freely moving was kept the same as possible for both tapered-tips in order for a fair comparison.

## 5. Conclusions and Outlook

In conclusion, we demonstrated a new type of acoustic vibration sensor based on a tapered-tip optical fiber microcantilever. These fibers exhibit a unique circular fringe pattern that changes dramatically with acoustic vibrations. Moreover, these tips resonate at distinct frequencies, which amplify the sensor response and thereby improve sensor sensitivity while reducing the detection limit down to the sub-nanometer range. Compared to a non-tapered-tip optical fiber, we observed a one order of magnitude better detection limit and two orders of magnitude higher sensitivity. Our results show that this type of acoustic vibration sensor can be a very good candidate for the PA detection of trace gases with very high sensitivity.

In our future work, we will try methane detection by placing this sensor in a gas chamber and using an excitation laser that will be matched to a vibrational mode of methane at λ = 1650.5 nm (due to the availability of this source in our lab). The laser source will be modulated at one of the resonance frequencies of the fiber sensor to amplify the detection sensitivity. Note that the minimum detection limit of methane is 5 ppm, which can be challenging to measure. On top of using highly sensitive sensors, one can also increase the acoustic field strength to reach this small detection limit. To do so, we developed resonator tubes [32], which can amplify the acoustic field strength by at least one to two orders of magnitudes.

## Figures and Tables

**Figure 1 sensors-23-00924-f001:**
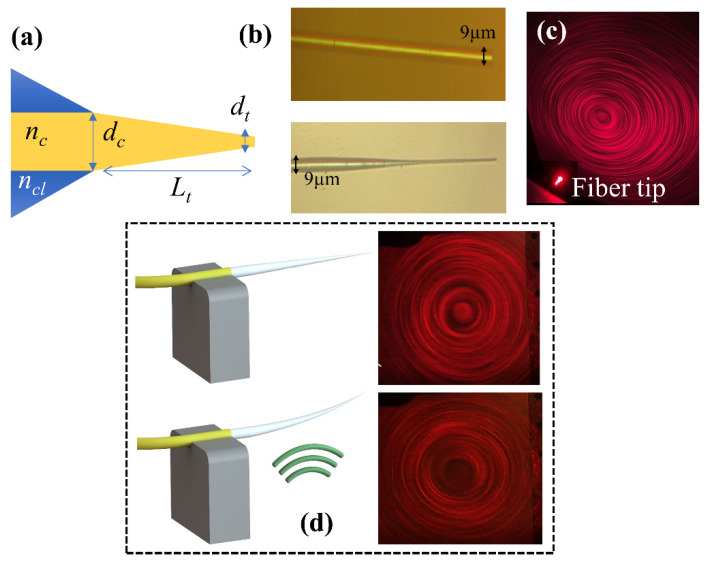
(**a**) Schematic of the tapered-tip fiber with relevant design parameters. (**b**) Microscope image of the fiber before etching (**top**) and after etching (**bottom**). (**c**) The output fringe pattern of the tapered-tip fiber in air, and the red spot is the tip. (**d**) The change in output fringe pattern upon acoustic field excitation. The top image is when the transducer is off, and the bottom image is when it is on.

**Figure 2 sensors-23-00924-f002:**
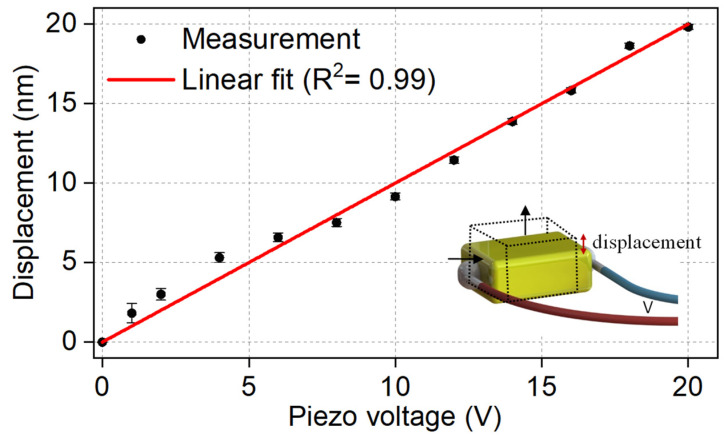
Measured displacement value of the piezo transducer at different voltage values. The inset shows the longitudinal displacement of the piezo when voltage is applied.

**Figure 3 sensors-23-00924-f003:**
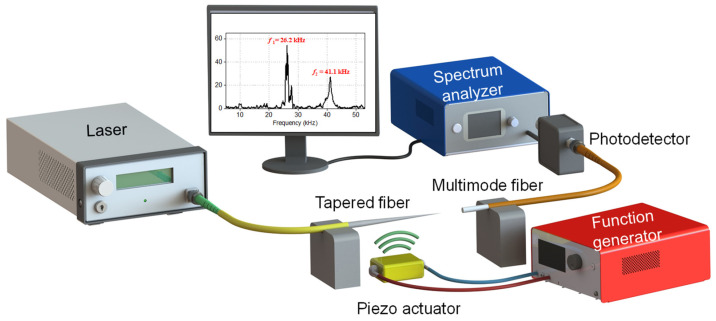
The schematic of the measurement setup. This is comprised of a single-wavelength laser centered at 1550 nm, a piezoelectric transducer, a function generator, tapered fiber, multimode fiber, a photodetector, and an electrical spectrum analyzer (ESA). The acoustic field generated by the piezo transducer vibrates the tapered-tip fiber, whose output is read by a multimode fiber and sent to the ESA via a photodetector.

**Figure 4 sensors-23-00924-f004:**
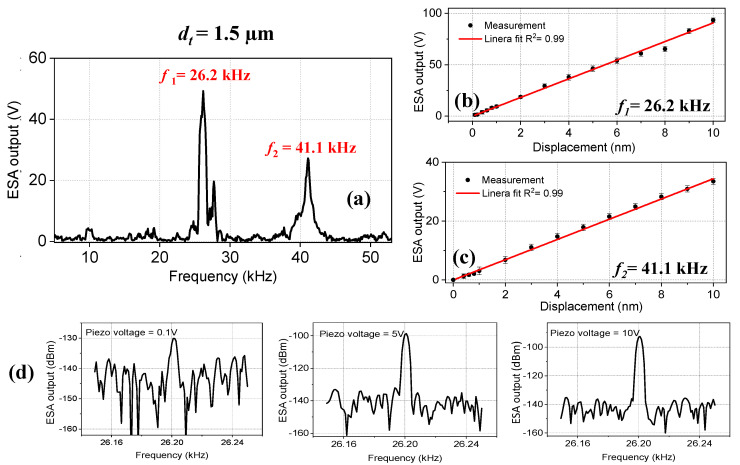
(**a**) The frequency spectrum of the tapered-tip fiber microcantilever with a tip diameter of *d_t_* = 1.5 µm. Two resonance frequencies were observed at *f*_1_ = 26.2 kHz and *f*_2_ = 41.1 kHz. The ESA output versus displacement values for (**b**) *f*_1_ = 26.2 kHz and (**c**) *f*_2_ = 41.1 kHz. The red line indicates the linear fit. (**d**) The ESA output spectrum at different *V* = 0.1 V, 5 V, and 10 V for *f*_1_ = 26.2 kHz.

**Figure 5 sensors-23-00924-f005:**
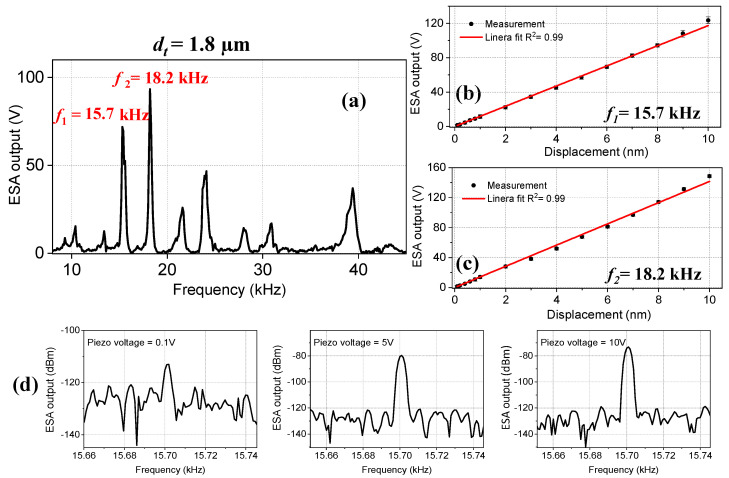
(**a**) The frequency spectrum of the tapered-tip fiber microcantilever with a tip diameter of *d_t_* = 1.8 µm. Several natural resonance frequencies were observed. The ESA output versus displacement values for the first two resonance frequencies of (**b**) *f*_1_ = 15.7 kHz and (**c**) *f*_2_ = 18.2 kHz. The red line indicates the linear fit. (**d**) The ESA output spectrum at different *V* = 0.1 V, 5 V, and 10 V for *f*_1_ = 15.7 kHz.

**Figure 6 sensors-23-00924-f006:**
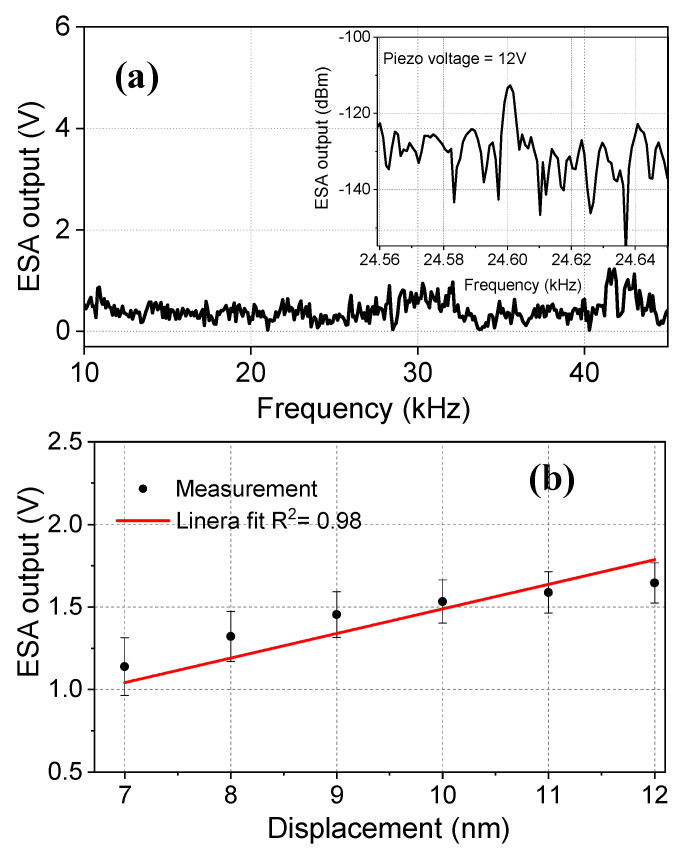
(**a**) The frequency spectrum of the regular fiber with a tip diameter of *d_t_* = 125 µm. No resonance was observed. The ESA output response at *f* = 24.6 kHz for the piezo voltage of 12 V is shown in the inset. (**b**) The ESA output versus displacement values at *f* = 24.6 kHz. The red line indicates the linear fit.

**Figure 7 sensors-23-00924-f007:**
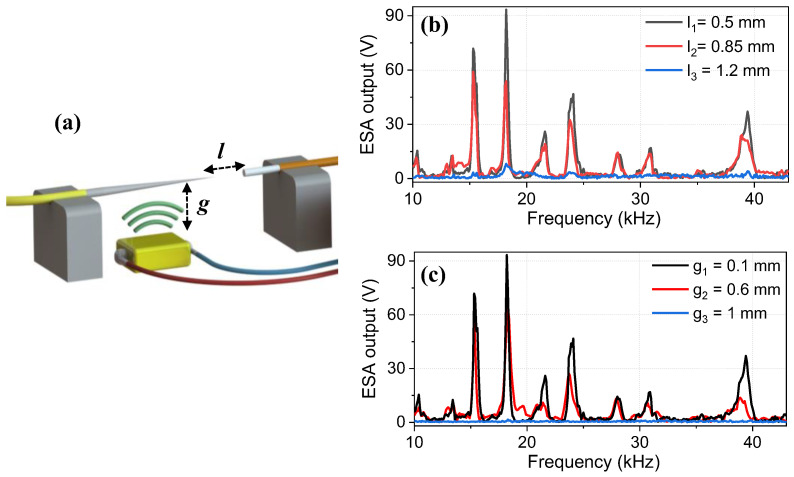
(**a**) Schematic indicating the parameters *l* and *g*. The effect of distance between (**b**) the tapered-tip fiber and the multimode fiber for three different values, and (**c**) the tapered-tip fiber and the piezo transducer for three different values on the frequency spectrum of the tapered-tip sensors. As can be seen, the frequencies of the resonances stayed the same but their amplitudes decreased with an increasing distance in both cases. Measurements were taken with tapered-tip fiber with a tip diameter of *d_t_* = 1.8 µm.

**Table 1 sensors-23-00924-t001:** Simulated eigenfrequencies of the tapered-tip fibers with different tip diameters.

Tip Diameter (µm)	Eigenfrequency (kHz)				
1.5	6.4	8.5	10.9	13.5	16.5
	19.7	23.2	27.2	31.2	35.9
	40.2	45.5			
1.8	5.9	7.7	10	12.4	15.0
	18.1	21.3	24.8	28.6	32.6
	37.2	41.6	46.5		

## Data Availability

The data presented in this study are available from the corresponding author upon reasonable request.

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
