# Peer review of "Sub-Nanometer Acoustic Vibration Sensing Using a Tapered-Tip Optical Fiber Microcantilever"

_sensors, 2023, doi:10.3390/s23020924_

Round 1

Reviewer 1 Report

Recommendation: Review Again After Resubmission (Paper is not acceptable in its current form, but has merit. A major rewrite is required. Author should be encouraged to resubmit a rewritten version after the changes suggested in the Comments section have been completed.)

Comments:

The authors demonstrate a highly sensitive acoustic vibration sensor that is based on a tapered-tip optical fiber acting as a microcantilever. The tapered-tip fiber has a unique output profile that exhibits a circular fringe pattern, whose distribution is highly sensitive to the vibration of the fiber tip. This is an interesting research topic. In my opinion some aspects need to be analyzed in the paper before the publication.

1. The working principle of this acoustic vibration sensor as a microcantilever should be discussed. The theoretical value of resonant frequencies of different tapered tip radius should be given and compared with the measured value.

2. The role of the length of the tapered-tip fiber is essential and should be discussed. The fabrication is in general well discussed. However, it is not reported how the authors select the length of the tapered-tip fiber. This length affect the sensitivity of the optical fiber sensor, also it affect the fringe pattern.

3. Each experimental result should be well discussed, including resonant frequencies, limit of detection, the sensitivity values, the differences between various piezo voltage, not just simply give figures or data.

4. Figure 2: all data point should be plot with error bar.

5. Line 122, page 4: ESA is not defined.

6. Figure3: Multimode fiber instead of Multiomode fiber.

7. Figure6: the unit of the inset of figure 6 a is a mistake. (dBm) instead of (V).

Reviewer 2 Report

After reviewing this manuscript, my comments are as follows.

1.     Please introduce the meaning of “IBM” (Line 33, Page 1) when the abbreviation appears in the manuscript for the first time.

2.     Please add a scale bar in Figure 1(b).

3.     As the authors claimed that “When an acoustic field is applied to the fiber tip, the tip starts bending periodically”. Can the authors provide a practical image of the fiber tip state when the acoustic field is applied?

4.     For Figure 2, please explain the parameter of “displacement” and add it in the schematic for a direct understanding. There is an error bar in Figure 2, so how many times did the authors measure?

5.     The fabrication process of the tapered fiber tip should be cleared.

6.     Please provide a microscope image of the fiber before etching.

7.     What’s the specific parameters (NA, core/cladding diameter, RI) of the MMF used in Figure 3?

8.     In the part of ‘Experimental Results’, the authors just listed the experimental results without any discussion and analysis.

9.     The meaning of “d” is not clear, for example, “d” is the radius of the fiber” (Line 79, Page 3) and is also the distance between the MMF and the fiber tip.

10.   The current discussion about the value of “d” and “g” is not valuable enough. It’s a common sense that the shorter distance will bring a better result. It’s better to provide the analysis about the minimum value of “d” and “g”.

11.   The authors previous reported paper entitled ‘Tapered tip optical fibers for measuring ultra-small refractive index changes with record high sensitivity’ Optics Letters 47(23):6281-6284 should be cited in this manuscript. In addition, the schematic diagrams of the tapered fiber tip proposed in the two works should be displayed differently.
